# ATP and Tri-Polyphosphate (TPP) Suppress Protein Aggregate Growth by a Supercharging Mechanism

**DOI:** 10.3390/biomedicines9111646

**Published:** 2021-11-09

**Authors:** Jordan Bye, Kiah Murray, Robin Curtis

**Affiliations:** Department of Chemical Engineering and Analytical Science, University of Manchester, Manchester M1 7DN, UK; jordanbye89@gmail.com (J.B.); murraykiah@gmail.com (K.M.)

**Keywords:** biopharmaceuticals, protein aggregation, protein–protein interactions, ATP, membraneless organelles, protein self assembly

## Abstract

A common strategy to increase aggregation resistance is through rational mutagenesis to supercharge proteins, which leads to high colloidal stability, but often has the undesirable effect of lowering conformational stability. We show this trade-off can be overcome by using small multivalent polyphosphate ions, adenosine triphosphate (ATP) and tripolyphosphate (TPP) as excipients. These ions are equally effective at suppressing aggregation of ovalbumin and bovine serum albumin (BSA) upon thermal stress as monitored by dynamic and static light scattering. Monomer loss kinetic studies, combined with measurements of native state protein–protein interactions and ζ-potentials, indicate the ions reduce aggregate growth by increasing the protein colloidal stability through binding and overcharging the protein. Out of three additional proteins studied, ribonuclease A (RNaseA), α-chymotrypsinogen (α-Cgn), and lysozyme, we only observed a reduction in aggregate growth for RNaseA, although overcharging by the poly-phosphate ions still occurs for lysozyme and α-Cgn. Because the salts do not alter protein conformational stability, using them as excipients could be a promising strategy for stabilizing biopharmaceuticals once the protein structural factors that determine whether multivalent ion binding will increase colloidal stability are better elucidated. Our findings also have biological implications. Recently, it has been proposed that ATP also plays an important role in maintaining intracellular biological condensates and preventing protein aggregation in densely packed cellular environments. We expect electrostatic interactions are a significant factor in determining the stabilizing ability of ATP towards maintaining proteins in non-dispersed states in vivo.

## 1. Introduction

Next-generation antibody therapeutics include antibody-drug conjugates, multispecific antibodies and antibody fusions are heavily engineered to meet biological efficacy, but often leading to poor stability and increased propensity to form aggregates, which are strictly regulated and need to be mitigated against [1,2,3,4,5]. One approach to offset the undesirable behaviour is to increase the protein colloidal stability through protein engineering or by adding co-solvents to the formulation. However, being successful requires an improved understanding of native-state protein–protein interactions and the relationship to colloidal stability, and in turn, understanding the role of colloidal stability in aggregation pathways.

Temperature-induced aggregation resistance can be substantially improved by modulating electrostatic interactions to increase the colloidal stability of the unfolded state. This has been achieved by developing positive and negative supercharged variants of proteins by selective mutagenesis of solvent-exposed regions to acidic or basic residues or by attaching acidic amino acids at the N-terminus of an IgG [6,7,8,9]. Supercharged variants are more resistant to heat-induced aggregation and most regain their native structure when cooled after heat treatment, whereas the wild-type proteins are much more likely to undergo irreversible aggregation. However, supercharging causes a decrease in conformational stability due to intramolecular electrostatic repulsion, which becomes more significant with an increase in the charged state [10,11]. This trade-off between colloidal stability and conformational stability also dictates the behaviour of many proteins, in particular mAbs, under acidic conditions [12,13,14,15,16]. At low pH and low ionic strength conditions, a large net positive charge causes a reduction in the unfolding free energy, but an increase in the colloidal stability. The increased colloidal stability predominantly impacts the aggregate growth mechanism leading to slower rates and causing the growth mechanism to switch from aggregate-aggregate coalescence to growth by monomer addition and then to nucleation dominated growth where aggregates form but do not grow [14,15,17,18,19,20]. However, under acidic conditions, although aggregate growth is suppressed, monomer loss is greatest due to increased rates of forming non-native aggregation-prone states [14].

Recently it was proposed that the small polyvalent anion adenosine triphosphate (ATP) plays a role in regulating the formation of membraneless organelles and suppressing protein aggregation and amyloid formation in the crowded cell environment [21,22,23,24,25]. In addition, low concentrations (10 mM) of ATP-Mg and its non-hydrolysable analogue APPCP-Mg prevent heat-induced aggregation of crude egg white and purified egg proteins [22]. The authors attributed the behaviour to the hydrotropic nature of ATP suggesting the adenosine group is required to interact with solvent-exposed hydrophobic regions of folded and/or unfolded proteins and the tri-phosphate moiety provides a protective layer of hydration. More recently, Mehringer et al. [26] demonstrated that the tri-polyphosphate (TPP) ion is equally effective as ATP at preventing aggregation of egg proteins and bovine serum albumin. The authors hypothesised the stabilizing mechanism might originate from the kosmotropic nature of phosphate groups, which would lead to preventing local exposure of protein sticky groups upon thermal stress.

In this study, we hypothesize that ATP suppresses protein aggregate growth through overcharging proteins causing an increase in electrostatic repulsion and colloidal stability. In contrast to overcharging proteins through mutagenesis, we expect that multivalent ion binding to proteins will not lead to the trade-off of poor conformational stability against the increase in colloidal stability. If this is the case, because ATP also binds to proteins non-specifically [23,27,28,29], an improved understanding of the electrostatic stabilization mechanism could lead to a rational strategy for controlling the behaviour for a broad range of protein types. Along these lines, we have previously shown that lysozyme phase behaviour can be tuned through non-specific binding of tripolyphosphate (TPP) [30]. At the same time, we hope to gain more insight into the biological role of ATP towards modulating protein assembly processes in vivo.

The aggregation behaviour for bovine serum albumin (BSA), ovalbumin, lysozyme, ribonuclease A (RNaseA), and α-chymotrypsinogen (α-Cgn) has been characterized in solutions with varying concentrations of ATP, sodium tripolyphosphate (STPP), or NaCl, which is used as a control for determining the effects of electrostatic screening. ζ-potential measurements have been used to quantify the extent of protein overcharging through ion binding, while temperature-ramped dynamic light scattering (DLS) and static light scattering (SLS) experiments have been used to evaluate how ions influence protein aggregation behaviour. We have also measured native-state protein–protein interactions in terms of osmotic second virial coefficients ($B_{22}$) by static light scattering, characterised the melting temperature ($T_{m}$) using differential scanning fluorimetry (DSF), and carried out monomer loss kinetics at elevated temperatures.

## 2. Materials and Methods

### 2.1. Materials

α-Cgn, BSA, lysozyme, and RNaseA were sourced from Sigma Aldrich (Gillingham, UK) with purities > 95% and ovalbumin was sourced from MP Biomedicals (Santa Ana, CA 92707) with a purity > 80%. Tris base, sodium chloride and adenosine triphosphate (ATP) were sourced from Sigma Aldrich (Gillingham, UK) with purities ≤ 99%. Sodium tripolyphosphate (STPP) was sourced from Fisher Scientific U.K. Ltd., (Loughborough, UK) with a purity of ≤99%. Water sourced from a Milli-Q^®^ Advantage A10^®^ water purification system (Merck, Darmstadt, Germany) with a resistivity of 18.2 MΩ.cm was used as the solvent for all salt and protein solutions.

All buffer solutions used for dialysis and excipient solutions were prepared volumetrically and filtered with a 0.2 µm hydrophilic nylon membrane (Merck Millipore Ltd., Tullagreen, Ireland). Ionic strength was calculated for each salt using $pK_{a}$ values obtained from the literature and the Henderson–Hasselbalch equation. A 10 mM tris buffer was used for all experiments.

### 2.2. Sample Preparation

Five millilitres of protein stock solution at 15 mg/mL was prepared and dialysed against 500 mL of the desired buffer for 4 h twice and again overnight at 4 °C. After dialysis, pH was checked and adjusted if necessary to reach the desired value. Protein stock solutions were passed through a 0.1 µm and then a 0.02 µm Whatman Anotop syringe filter (Scientific Laboratory Supplies Ltd., Nottingham, UK) and stored on ice. The filtered stock solutions were used in all the experiments.

A special note on the preparation of ovalbumin should be made. Dynamic light scattering (DLS) measurements revealed the presence of large aggregates with radii around 100 nm. Prior to dialysis, the aggregates were removed by preparing a dilute ovalbumin solution (2 mg/mL) in 10 mM tris pH 7.0 buffer, then vacuum filtering the protein solution with a 0.1 µm Durapore^®^ membrane filter (Merck Millipore Ltd., Tullagreen, Ireland) and then a 0.025 µm Durapore^®^ membrane filter (Merck Millipore Ltd., Tullagreen, Ireland). After filtration, the sample was concentrated to 15 mg/mL using a Sartorius 10 kDa MWCO Vivaspin 20 centrifugal concentration unit (Sigma Aldrich, Gillingham, UK). The protein solution was then dialysed and filtered as described in the paragraph above.

### 2.3. ζ-Potential Measurements

ζ-potentials for ovalbumin and α-Cgn were determined on a Zetasizer Nano ZSP (Malvern Instruments Ltd., Malvern, UK) using DTS1070 folded capillary cells (Malvern Instruments Ltd., Malvern, UK). All ζ-potential measurements were made with 1 mg/mL protein concentration at 25 °C. The same applied voltage (150 V) was used for all measurements. Henry’s function was set equal to 1.5 according to the Smoluchowski approximation. The sample was allowed to equilibrate for 120 s before 10 measurements were were collected and averaged. Each sample condition was repeated 3–6 times. Error bars correspond to the standard deviation across replicate measurements, and in some instances, appear smaller than the datapoints.

### 2.4. Temperature Ramped Dynamic Light Scattering (DLS and SLS)

Thermal ramps and isothermal experiments at high temperatures were carried out on a Wyatt DynaPro NanoStar (Wyatt Technology Corporation, Santa Barbara, CA, USA), using a laser wavelength of 658 nm, with separate DLS and SLS detectors located at 90° to the incident laser light. A sample at a protein concentration of 10 mg/mL and the target salt concentration was prepared and passed through a 0.02 µm Whatman Anotop syringe filter (Scientific Laboratory Supplies Ltd., Nottingham, UK). For each experiment, 100 µL of the sample was loaded into a Wyatt 1 µL Quartz Cuvette (Wyatt Technology Corporation, Santa Barbara, CA, USA). For thermal ramp experiments, the cuvette was allowed to equilibrate at the starting temperature for 600 s, then heated at 1 °C/min from 30–80 °C. If significant aggregation occurred before reaching the upper-temperature limit (80 °C), the scan was stopped prematurely to prevent excessive fouling of the cuvette. For isothermal runs, the loaded cuvette was placed in the cuvette holder after the sample chamber had reached the target temperature and allowed to equilibrate for 300 s before taking any readings. For all runs, the acquisition time was set at 6 s, which corresponds to a DLS acquisition being collected every 0.1 °C throughout the thermal ramp scans. Each sample condition was run in duplicate unless otherwise noted.

Intensity autocorrelation functions were analysed using the DYNAMICS software (Wyatt Technology Corporation, Santa Barbara, CA, USA). Fits to the correlation function were performed between 1.5 and 6 × 10^4^ µs using a cumulant analysis and regularization analysis implemented by the DYNAMICS software. The cumulant analysis was used to determine the *z*-average hydrodynamic size $R_{H}$ and the polydispersity $P_{d}$, which is related to the gaussian width of decay rates. For ovalbumin and RNaseA samples, a two-decay model was fit to the intensity autocorrelation function to separate out the contributions from the monomer and the aggregate. The error bars correspond to the standard deviation across the replicate measurements.

### 2.5. Temperature Ramped Thermal Stability to Determine T_m_

Protein melting temperatures were measured by monitoring fluorescence as a function of temperature using an UNcle instrument (Unchained Laboratories, Pleasanton, CA, USA), which contains a laser with 266 nm wavelength to excite samples. Samples were equilibrated at 30 °C for 5 min before measurements. A 10 μL sample with 1 mg/mL protein and varying salt concentration was loaded into each microcuvette. Unfolding experiments were carried out over 30–90 °C with a scan rate of 1 °C min^−1^. All measurements were taken in triplicate. A higher *T*_m_ value when changing solution conditions generally indicates the protein structure is being stabilised. The error bars reported in the results section correspond to the standard deviation across the triplicate measurements.

### 2.6. Size Exclusion Chromatography Multiangle Laser Light Scattering (SEC-MALLS)

1 mL samples of 1 mg/mL ovalbumin with the desired STPP or NaCl concentration in 10 mM Tris pH 7.0 buffer were prepared. These samples were passed through a 0.02 µm Anotop filter membrane and incubated at 70 °C in a water bath for 0, 5, 10, 15, 20 and 30 min before being removed and placed on ice. Samples were then loaded into an autosampler at 4 °C, which was set to inject 100 µL of sample.

Size-exclusion chromatography coupled with multi-angle laser light scattering (SEC-MALLS) was performed using an Agilent 1100 Series HPLC (Agilent Technologies, Waldbronn, Germany) with a Wyatt DAWN EOS (Wyatt Technology Corporation, Santa Barbara, CA, USA). The Agilent 1110 Series HPLC has a four-line binary pump system, degasser, temperature-controlled auto-sampler (4–40 °C) and 0.6 cm cell Diode Array Detector (DAD). Protein separation was performed with a TSKgel G3000SWxl column with a 0.5 µm pre-filter (Tosoh Bioscience LLC, King of Prussia, PA, USA). The aqueous mobile phase was filtered with a 0.1 µm Durapore^®^ membrane filter (Merck Millipore Ltd., Tullagreen, Ireland) and consisted of the corresponding TPP or NaCl concentration in 10 mM Tris at pH 7.0, where a flow rate of 1 mL/min was used. The Wyatt DAWN EOS is an 18-angle (15–160°) static light scattering (SLS) instrument. It uses a GaAs laser with a wavelength of 685 nm. The UV signal acquired at 280 nm was simultaneously channelled to the analogue input on the Wyatt DAWN EOS. The Wyatt Astra 6.1 software (Wyatt Technology Corporation, Santa Barbara, CA, USA) uses the UV and SLS signals to calculate molecular weights of eluted species.

### 2.7. $B_{22}$ Determination by SLS

SLS experiments were conducted on a Wyatt miniDAWN TREOS 3 angle (49°, 90° and 131°) detector (Wyatt Technology Corporation, Santa Barbara, CA, USA) and the Wyatt Calypso II (Wyatt Technology Corporation, Santa Barbara, CA, USA) was used for automated syringe delivery of the samples. The method and analysis of the SLS data for obtaining $B_{22}$ values have been previously described [30,31]. The measurements require a value for the protein refractive index increment, which was set equal to 0.186 mL/g for solutions with ovalbumin or α-Cgn. The molecular weights obtained from extrapolating light scattering data to zero protein concentration agreed within ±1 kDa of the protein monomer sequence molecular weight (42.7 kDa for ovalbumin and 25.6 kDa for α-Cgn). At salt concentrations of either ATP or STPP greater than 5 mM, the measured molecular weights decrease below monomer value when using 0.186 mL/g for the refractive index increment. For these runs, the increment was fit to obtain the monomer molecular weight following the procedure described in Holloway et al. [32]. Results are reported in terms of a reduced osmotic second virial coefficient given by $b_{22}=B_{22}/B_{22}^{\mathrm{hs}}$ where the hard-sphere contribution in volume units is estimated using $B_{22}^{\mathrm{hs}}=\left(16/3\right)\pi R_{H}^{3}$ [33].

## 3. Results

### 3.1. ζ-Potential Measurements

ζ-potential measurements shown in Figure 1 were used to determine how NaCl and STPP influence the net charge of ovalbumin and α-Cgn at pH 7.0. The ζ-potential indicates the electrostatic potential at the slip plane of the protein, where strongly bound counterions and co-ions are contained within the surface of hydrodynamic shear. ζ-potential values of the two proteins tested here at zero salt concentration reflect their net charge at pH 7.0. The measured isoelectric pH value is 4.6 for ovalbumin, which is expected to have a net negative charge in the absence of any ion binding, while a slight positive charge is expected for α-Cgn, which has a pI around 8.3. The ζ-potential measurements for NaCl do not change much with increasing ionic strength. On the other hand, increasing STPP concentration causes the ζ-potential for both proteins to become markedly more negative indicating both proteins are overcharged by TPP binding. Overcharging by TPP has been previously observed for lysozyme [30] and an intrinsically disordered protein histatin-5 [27], while similar effects occur for solutions of acidic proteins with salts of trivalent cations [34]. In these previous studies, the sign of the multivalent ion charge is opposite to the protein net charge sign. Increasing salt concentration initially leads to protein charge neutralization followed by the overcharging effect at higher salt concentration. In contrast, in our study, overcharging effects occur at much lower salt concentration, because α-Cgn is close to its pI and the net charge sign of ovalbumin is the same as TPP.

### 3.2. Thermally Induced Aggregation Experiments

Simultaneous DLS and SLS measurements were made as a function of temperature for solutions containing either ovalbumin, α-Cgn, or BSA with different concentrations of NaCl, STPP and ATP. A cumulant analysis was used to determine temperature profiles of $R_{H}$, which are shown for ovalbumin (see Figure 2), BSA (see Figure 3a), and α-Cgn (see Figure 3b) for selected solution conditions (see Appendix A of the SI for data at all solution conditions). The $R_{H}$ of the three proteins remains constant at low temperatures indicating that the proteins remain in their native states. For all runs at low temperature, $R_{H}$ values of 2.9 nm, 2.5 nm, and 3.8 nm were recorded for ovalbumin, α-Cgn and BSA, which agree with reported literature values [35,36,37]. Increasing NaCl concentration causes the aggregation to shift to higher temperatures for α-Cgn and BSA (see Figure 3a,b, respectively), whereas the addition of NaCl to ovalbumin causes aggregation to occur more at lower temperatures (see Figure 2c).

The effects on ovalbumin aggregation of STPP and ATP follow the same pattern. In solutions with STPP or ATP, aggregation is shifted to a higher temperature and the aggregation suppression effects are always greater than for NaCl at equivalent molar concentrations. The $R_{H}\;$profiles become less curved upon an initial increase in STPP or ATP concentration, but the dependence is non-monotonic, where minimum curvature occurs over a concentration range of 5 to 25 mM. Above a concentration of 25 mM, the curvature remains less than the additive-free solution. Figure 2d shows that the profiles at salt concentrations of either 10 mM or 50 mM are almost identical for ATP and STPP. The only discrepancy between ATP and STPP occurs at a salt concentration of 0.5 mM, where aggregation relative to the buffer solution is enhanced with ATP but suppressed by STPP.

A similar pattern of behaviour occurs for BSA and for ovalbumin in solution with STPP (compare Figure 2a and the inset to Figure 3a). For both proteins, increasing the concentration of STPP to 10 mM reduces the curvature in the $R_{H\;}$profile. For BSA, further increasing the salt concentration increases the curvature, but to a lesser extent than observed with ovalbumin. While we have not measured solutions of BSA with ATP, a recent study found the aggregation behaviour upon thermal stress as a function of salt concentration is identical for ATP and STPP at pH 7.4 with a 50 mM Tris buffer [26].

The $R_{H}$ profiles observed with α-Cgn shown in Figure 3b follow a different pattern when compared with ovalbumin or BSA. For all salts, increasing their concentration shifts the $R_{H}$ profiles to higher temperatures. However, in contrast to BSA and ovalbumin in solutions with STPP, for α-Cgn samples, there is always a rapid increase in $R_{H}$ above the aggregation onset temperature (the temperature where an initial change in $R_{H}$ is detected). The inset to Figure 3b contains the measured melting temperatures $T_{m}$. Increasing STPP concentration causes a similar increase in the $T_{m}$ and the aggregation onset temperature, which indicates the main effect of STPP on aggregate suppression is through stabilizing α-Cgn against unfolding. Because the slopes of the $R_{H}$ profiles are slightly greater for STPP containing solutions, we expect the TPP ions are not slowing down aggregate growth. The aggregation onset temperature is shifted upwards by 1 °C for solutions with 100 mM ATP versus with 100 mM STPP, while similar onset temperatures are observed in solutions with either 50 mM STPP or 50 mM ATP (see Appendix A).

### 3.3. Analysis of Ovalbumin Thermal Ramp Data Suggests STPP Alters Aggregate Growth Rates

The initial increase in $R_{H}$ with increasing temperature is due to aggregate formation and growth. To gain further insight into how the salts alter the rates of these steps, we have fit the electric-field correlation function to a two-decay model given by
$$g^{\left(1\right)}\left(\tau \right)=f_{\mathrm{mon}}\exp\left(-D_{\mathrm{mon}}q^{2}\tau \right)+f_{\mathrm{agg}}\exp\left(-D_{\mathrm{agg}}q^{2}\tau \right)$$
where τ is the delay time, q is the magnitude of the scattering vector, $f_{\mathrm{mon}}$ and $f_{\mathrm{agg}}$ are the fractions of the light scattered by the monomer and the aggregate, respectively, and $D_{\mathrm{mon}}$ and $D_{\mathrm{agg}}$ are the diffusion coefficients of the monomer and the aggregate, respectively, which are related through the Stokes–Einstein relation to the apparent hydrodynamic sizes, $R_{H,\mathrm{mon}}$ and $R_{H,\mathrm{agg}}$, respectively. The accuracy of the fits as characterised in terms of $\chi^{2}$ values are comparable to the cumulant analysis for the conditions when the ratio $R_{H,\mathrm{agg}}/R_{H,\mathrm{mon}}<5$. Similar $\chi^{2}$ values are obtained for larger ratios, but the cumulant analysis is no longer accurate since the population becomes multi-nodal.

In order to check that the fitting parameters have physically realistic values, we have compared the fit value of $f_{\mathrm{mon}}$ against the calculation from the static light scattering reading, where $f_{\mathrm{mon}}\left(T\right)=I_{\mathrm{mon}}^{\mathrm{ex}}/I^{\mathrm{ex}}\left(T\right)$. Here $I_{\mathrm{mon}}^{\mathrm{ex}}$ is the excess light scattering intensity of the sample before aggregation, and $I^{\mathrm{ex}}\left(T\right)$ corresponds to the intensity as a function of temperature above the onset of aggregation. The comparison shown in Figure 4a indicates there is good agreement between the fitting results (shown by symbols) and the calculations based on the SLS signal (shown by lines). For the run with 10 mM STPP, it is not possible to obtain accurate estimates for the fit parameters when $f_{\mathrm{mon}}\;$is small due to strong correlations between the fitting parameters that arise when $R_{H,\mathrm{agg}}$ is only slightly greater than $R_{H,\mathrm{mon}}$. It should be noted that $f_{\mathrm{mon}}$ is roughly proportional to the inverse of the SLS intensity, which is why it follows the same pattern as the $R_{H}$ profiles with respect to the solution conditions. The greatest shift to higher temperatures occurs for the conditions with 10 mM STPP, while the conditions with 100 mM NaCl are shifted to lower temperatures.

The value of $R_{H,\mathrm{agg}}$ is a more direct measure of the average aggregate size since $R_{H}$ corresponds to the average including the monomer population. The fit values of $R_{H,\mathrm{agg}}$ shown in Figure 4c as a function of temperature can be used to rationalize why there is a peak in the $P_{d}$ profiles shown in Figure 4b for the conditions with STPP. The initial increase of $P_{d}$ with temperature occurs along with the concomitant increase in when there is still a significant fraction of the light being scattered by the monomer. The increase arises because the monomer and the aggregate are contributing to the light scattering signal. The peak in $P_{d}\;$occurs at a temperature where $f_{\mathrm{mon}}\approx 0.3$. With further increasing temperature $P_{d}\;$reduces since the light scattering intensity becomes dominated by the aggregates. When $f_{\mathrm{mon}}\to 0$, $P_{d}\;$reflects the width of the aggregate size distribution, which remains low for the condition with 5 to 25 mM STPP (data only shown for 10 mM STPP) but does increase for the other STPP conditions at the higher temperatures. This provides some evidence that there could be a cross over in the aggregate growth mechanism when altering the salt concentration. A low polydispersity of the aggregate distribution is a signature of growth by chain polymerization, which is most pronounced at intermediate STPP concentrations, while an increased polydispersity reflects growth by aggregate-aggregate coalescence [15,38], which is most evident in the absence of STPP. For solutions with 1 and 100 mM STPP, the increase in $P_{d}$ above 76 °C likely indicates there is some aggregate-aggregate coalescence.

The results of fitting to the two-decay model can also be used for estimating the fraction of the protein that is aggregated. The excess normalised scattered light intensity $I^{\mathrm{ex}}$ from the aggregate is given by $f_{\mathrm{agg}}I^{\mathrm{ex}}\sim c_{\mathrm{agg}}M_{\mathrm{agg}}$ where $c_{\mathrm{agg}}$ and $M_{\mathrm{agg}}$ are the mass concentration of aggregated protein and weight average aggregate molecular weight, which, in turn, can be approximated from the fit value for the aggregate radius $R_{H,\mathrm{agg}}$ according to $M_{\mathrm{agg}}\sim R_{H,\mathrm{agg}}^{d_{f}}$, where $d_{f}$ is the fractal dimension of the aggregate. This approach is only approximate as $d_{f}$ depends upon a number of parameters such as the time evolution of the aggregate, as well as the strength of the electrostatic interactions. The results are not very sensitive to the choice of $d_{f}$, which is set equal to 1.8 based on literature studies of ovalbumin [39]. Figure 4d contains plots of $f_{\mathrm{agg}}I^{\mathrm{ex}}/R_{H,\mathrm{agg}}^{d_{f}}\sim c_{\mathrm{agg}}$ which is also expected to be a measure of the fraction of monomer loss since $c_{\mathrm{agg}}=c-c_{\mathrm{mon}}$ where $c_{\mathrm{mon}}$ is the mass concentration of the unaggregated protein and c is the concentration used in the experiment. For the runs with 10 mM STPP, reliable results are only obtainable once the temperature is greater than ~72.5 °C. All the profiles have a similar shape but are offset from each other with respect to changes in temperature. In particular, the curves for 1 mM and 10 mM STPP overlay with each other, even though the corresponding $R_{H}$ profiles differ considerably. This suggests the monomer loss kinetics are similar to each other, but the aggregate growth is reduced at 10 mM versus 1 mM STPP.

In order to provide a qualitative comparison of monomer loss rates across different solution conditions, we have defined a monomer loss temperature $T_{\mathrm{mon}}$ as the temperature where $f_{\mathrm{agg}}I^{\mathrm{ex}}/R_{H,\mathrm{agg}}^{d_{f}}=0.006$ nm^−1.8^ (see Figure 5a). Solutions containing STPP or ATP over a concentration range of 1 to 10 mM exhibit similar amounts of monomer loss as the values of $T_{\mathrm{mon}}$ remain relatively constant, although a slight increase of 2 to 3 °C occurs when increasing the salt concentration up to 100 mM. Previous studies on ovalbumin and charged mutants of ovalbumin indicate monomer loss kinetics under thermal stress correlate with the protein melting temperature $T_{m}$ [40]. In Figure 5a, the measured values of $T_{m}$ are shown for STPP and for NaCl solutions. For NaCl, there is little variation in $T_{m}$ (less than ±1 °C) which is consistent with the small variation in the corresponding $T_{\mathrm{mon}}$ values. Similarly, for STPP solutions, the values of $T_{m}$ parallel the changes in $T_{\mathrm{mon}}$. While we have not measured $T_{m}$ for STPP concentrations below 10 mM, Mehringer et al. [26] showed there is only a slight increase of $T_{m}$ in solutions containing 1 mM STPP relative to a pH 7.4 buffer solution, while the values remain constant between 1 mM and 40 mM.

A relative measure of the aggregate growth rates is given by the aggregate size compared across conditions where there is a similar extent of monomer loss. Figure 5b contains a plot of $R_{H,\mathrm{agg}}$ determined at the monomer loss temperature $T_{\mathrm{mon}}$. With increasing NaCl concentration, there is a dramatic increase in the $R_{H,\mathrm{agg}}$ values reflecting increased aggregate growth rates. On the other hand, there is a well-defined minimum in the aggregate growth rate as a function of TPP or ATP concentration. The behaviour observed with NaCl follows a similar pattern already observed in the literature, where increasing NaCl concentration does not alter monomer loss kinetics or conformational stability, but causes the formation of larger aggregates, an effect which has been attributed to screening of electrostatic repulsion [41,42]. Analogously, a similar finding has been drawn from aggregation studies of ovalbumin mutants with varying charged states between −1 and −26 *e*. The more highly charged mutants form smaller aggregates under low ionic strength conditions, but under high salt conditions, aggregates become larger and similar-sized to those formed by the low net-charge mutants [43]. As such, these studies suggest the minimum in aggregate growth rates observed with ATP or TPP should be rationalizable in terms of how the multivalent ions alter the electrostatic interactions between the aggregating proteins.

There is only a measurable effect of changing from STPP to ATP at a salt concentration of 0.5 mM. For samples with 0.5 mM ATP, the aggregate growth rates and the monomer loss kinetics appear to increase relative to the buffer solution. We currently do not have an explanation for this behaviour, which is opposite to what happens for all other ATP or STPP-containing solutions.

### 3.4. SEC-MALLS Measurements Indicate TPP Suppresses Aggregate Growth

We also carried out monomer loss kinetics during isothermal holds at 70 °C in a solution containing either NaCl or STPP to provide further support to the aggregate growth rates deduced from the thermal ramp experiments. Appendix A contains the chromatograms for the samples heated for 30 min. For each of the runs with STPP, the chromatograms also contain a peak corresponding to aggregates eluting from the column. For these samples, the total recovered protein calculated from integrating the aggregate and monomer peak areas is within 99% of the injected mass indicating negligible amounts of large aggregates in these samples. For the runs with NaCl, an aggregate peak is not observed because the aggregated protein is so large that it has been removed by the in-line filter. Figure 6 contains the monomer fraction and the weight-average molecular weight of the aggregate peak plotted versus time. The observations that monomer loss is insensitive to increasing STPP concentration, while aggregate size increases, indicates a concomitant increase in aggregate growth rates. The decreasing aggregate size observed when using STPP versus NaCl, or by reducing STPP concentration, is consistent with the trend of aggregate growth rates deduced from the thermal ramp experiments.

### 3.5. RNase A and Lysozyme

We have also used dynamic light scattering to measure the effect of STPP on the isothermal aggregation of RNAseA at 75 °C and pH 7 (10 mM tris buffer) and lysozyme at 70 °C and pH 8.5 (10 mM tris buffer), where the corresponding plots are shown in Figure 7a,b, respectively. The higher pH has been used for the lysozyme experiment to lower the temperature required to induce aggregation. RNaseA and lysozyme are expected to carry a net positive charge under the experimental conditions as their pI values are equal to 8.6 and 11.3, respectively. Lysozyme experiments could only be carried out at STPP concentrations greater than 30 mM as the protein was precipitated at lower concentrations. At 30 mM, the $R_{H}$ profile shifts to a lower temperature than for the buffer solution, while increasing STPP concentrations above this value shifts the curves to higher temperatures. However, the slopes of the profiles are similar to each other, so we do not suspect that there is a strong effect on aggregate growth. On the other hand, for RNaseA, a drastic change occurs with increasing the STPP concentration from 10 mM to 25 mM. The $R_{H\;}$ profiles for 25 mM STPP and for 50 mM NaCl are similar to each other. However, when the two-decay model is applied to the data, we find that the aggregate size for a given $R_{H}$ value is much greater in the NaCl solutions (see inset to Figure 7a), which indicates a lower aggregate growth rate in the STPP solutions.

### 3.6. $b_{22}$ Values Indicate ATP/TPP Overcharging Increases Protein–Protein Electrostatic Repulsion for Ovalbumin, but Not for α-Cgn

To get a better understanding of how the multivalent ions alter the electrostatic interactions between proteins, we have determined the reduced osmotic second virial coefficients $b_{22}$ for solutions of either ovalbumin or α-Cgn. Studies focused on elucidating aggregation pathways indicate $b_{22}$ values show a strong correlation with the aggregate growth mechanism for a number of mAbs and α-Cgn when varying pH and salt concentration under low ionic strength conditions [14,15,17]. $b_{22}$ measurements, carried out at room temperature, reflect native-state protein–protein interactions, while aggregate growth likely occurs through the addition of growth units containing proteins in partially folded states. The strong correlation between $b_{22}$ and aggregate growth mechanism provides an indication that the electrostatic interactions between native proteins are similar to those between aggregate growth units, which might not be surprising since most charged groups occur on protein surfaces.

Figure 8 shows a plot of the reduced osmotic second virial coefficients $b_{22}$ values for ovalbumin and for α-Cgn under the same solution conditions used for monitoring aggregation upon thermal stress. The independent variable is chosen to be ionic strength, which determines the range of electrostatic interactions in terms of the Debye screening parameter, which at room temperature is given by $\kappa \left(\mathrm{nm}\right)=3.29\sqrt{IS\left(M\right)}$. At pH 7, the ionic strength of ATP or STPP solutions is a factor of 9.07 or 9.96 times greater than the salt concentration, respectively, using the $pK_{a}$ values shown in Appendix A [44,45].

The behaviour for ovalbumin follows the typical behaviour of proteins in solutions a few pH units either below or above the pI, where increasing ionic strength over the range of 10 to 100 mM screens electrostatic repulsion [31,46,47,48,49,50]. Once the repulsion is sufficiently screened at higher ionic strengths, the $b_{22}\;$values are less than 1 reflecting the presence of attractive protein–protein interactions. When changing the salt from NaCl to STPP, the increase in $b_{22}$ at fixed ionic strength suggests there is an increase in the magnitude of the electrostatic repulsion due to overcharging the protein surface by binding multivalent anions, which has also been deduced from the ζ-potential measurements. The increased protein–protein repulsion observed in ATP versus STPP solutions exists even at high ionic strength where electrostatic forces are sufficiently screened. As such, it is unclear if ATP induces a stronger electrostatic repulsion or attenuates the attractive interactions between proteins, or there is a combination of these two effects.

On the other hand, for α-Cgn, $b_{22}$ values remain less than 0 for all solution conditions reflecting the absence of any electrostatic repulsion, which, for NaCl, is not surprising since the ζ-potential is relatively small in NaCl solutions. The non-monotonic dependence of $b_{22}$ arises due to the orientational anisotropy of the protein–protein interactions. Computational studies have shown the main contributions to $b_{22}$ arise from orientationally constrained configurations stabilised by protein surfaces with high geometric complementarity, where the electrostatic interactions are favourable [50,51,52]. The increase in $b_{22}$ at low ionic strength arises from screening these electrostatic interactions. Although ζ-potential measurements indicate STPP and ATP overcharge α-Cgn, the $b_{22}$ values are very similar to the corresponding ones with NaCl when plotted versus ionic strength. This trend suggests the main effect of STPP is to act as a screening salt and there is no impact of overcharging on the electrostatic repulsion. The lack of any electrostatic repulsion induced by ATP or TPP might explain why there is no observed aggregate growth suppression in the polyphosphate-containing solutions. When α-Cgn is thermally stressed under low pH conditions, the transition to aggregate growth by chain polymerization only occurs for conditions where $b_{22}>8$ [15].

### 3.7. Aggregate Growth Rates Show Stronger Correlation with FUCHS Factor Than $b_{22}$

The reduction in aggregate growth rates for ovalbumin occur under conditions where there is strong electrostatic protein–protein repulsion, but it is not clear why the values of $b_{22}$ correlate with aggregate growth rates for NaCl solutions, but not for solutions with either STPP or ATP. The well-defined minimum in aggregate growth observed for ATP or STPP is not reproduced by the $b_{22}\;$values. Increasing ATP or TPP concentration to 5 or 10 mM (or equivalently an ionic strength of 50 to 100 mM) has a dramatic effect on reducing aggregate growth, but the corresponding $b_{22}$ values are close to 1, which are much less than the values in the salt-free solution.

Protein aggregate growth rates are more directly related to the FUCHS factor, which accounts for the effect of the repulsive barrier in the protein–protein interaction potential under reaction limited cluster association (RLCA) conditions [20,53,54,55,56]. The repulsive barrier slows down the diffusion of primary aggregate units as they collide, which leads to a reduction in aggregate growth. For isotropic colloids, in the absence of a repulsive barrier, every collision step leads to aggregate growth. In this case, the inverse of the FUCHS ratio can be thought of as a sticking probability equal to the fraction of collisions in the presence of the repulsive barrier normalised by the number of collisions that would have happened in the absence of a barrier under diffusion-limited cluster association conditions. For interaction potentials with a range shorter than the size of a monomer, the aggregating unit can be approximated as the monomer protein and the FUCHS ratio is then given by:(1)$$W_{FUCHS}=2a\int_{2a}^{\infty }exp\left(\beta u\right)\frac{dr}{r^{2}}\;$$
where *β* is the inverse reduced temperature $1/k_{B}T$, where *T* is temperature and $k_{B}$ is Boltzmann’s constant, *r* is the protein centre-to-centre separation, and *a* is the effective spherical radius of the protein, which can be approximated by the hydrodynamic radius (equal to 2.9 nm for ovalbumin). As long as the aggregate morphology remains invariant, the FUCHS factor is expected to be proportional to the timescale for aggregates to coalesce with each other assuming the primary aggregating unit can be described by a native protein.

Calculating the FUCHS factor requires fitting a model for the protein–protein interaction potential to the measured $b_{22}\;$values. The electrostatic contribution to the interaction potential $u^{\mathrm{el}}$ for charged proteins under low ionic strength conditions can be adequately described by the double-layer potential from Derjaguin–Landau–Verwey–Overbeek (DLVO) theory:(2)$$\beta u^{\mathrm{el}}\left(r\right)=\frac{Z^{2}l_{B}}{{\left(1+\kappa a\right)}^{2}}\frac{\exp\left[-\kappa \left(r-2a\right)\right]}{r}$$
where $l_{B}$ is the Bjerrum length, *Z* is the protein net charge. In addition, there are other attractive contributions to the interaction potential that are not well defined due to the heterogeneous nature of the protein surface. However, the attractive forces tend to be short-ranged, in which case the virial coefficient can be approximated by $b_{22}=1+b_{22}^{\mathrm{el}}+b_{22}^{\mathrm{att}}$ [31], where $b_{22}^{\mathrm{att}}$ corresponds to the contributions from attractive interactions, and the electrostatic term is given by:(3)$$b_{22}^{el}=\frac{3}{8a^{3}}\int_{2a}^{\infty }\left[1-exp\left(-\beta u^{el}\right)\right]r^{2}dr$$

The only fitting parameters to the model are the net attractive contribution given by $b_{22}^{\mathrm{att}}$ and the magnitude of the protein net charge *Z*, which is assumed to be independent of salt concentration. In Figure 8a, we show the model calculations for the NaCl data, assuming that *Z* is a constant obtained from fitting to the data. The fit value equal to 11.8 *e* agrees very well with values calculated from the potentiometric titration [57], which provides an indication the model is able to capture the electrostatic contribution to the interaction potential.

A similar approach can be applied to the data obtained for solutions with either ATP or TPP except the value for *Z* will reflect the change in protein charge due to ion binding. As such, we have fit Z to match each $b_{22}$ value, where the results are presented in Figure 8c. With increasing salt concentration, the charge parameter remains relatively constant as expected for NaCl solutions, but there is an increase in *Z* for both ATP and TPP solutions reflecting an increase in ion binding. The fit values for ATP solutions are largest reflecting the increased $b_{22}$ values.

There is a strong correlation of protein aggregate growth rates with $W_{\mathrm{FUCHS}}$ reflecting the significance of electrostatic interactions. Figure 8d contains the values for $W_{\mathrm{FUCHS}}$ calculated using the fit values of *Z*, where increasing values of $W_{\mathrm{FUCHS}}$ should correlate with decreased aggregate growth rates. With increasing concentration of ATP or TPP, there is an increase in $W_{\mathrm{FUCHS}}$. For ATP, the minimum aggregate growth is predicted to occur above a salt concentration of 1 mM (or ionic strength of about 20 mM). On the other hand, for STPP solutions, the increase in the FUCHS factor is mainly observed at lower salt concentrations of 1 to 2.5 mM (ionic strengths of 20 to 36 mM), but the values at 5 to 10 mM STPP concentration are reduced, although these correspond to the minimum in the aggregate growth rates. Our analysis relies on the applicability of the double layer potential from DLVO theory to describe the longer-ranged electrostatic interactions between proteins. However, the potential is derived using the Poisson–Boltzmann equation, which is a mean-field theory where the ions are treated as an ideal gas. The approximation works well for salts of monovalent ions but breaks down for multivalent ions due to strong ion–ion correlations [58,59]. In particular, for asymmetric salts, the screening length $\kappa^{-1}$ is over-estimated when using mean-field theories. In Figure 8a,d, we also show the results of fitting the double layer potential using a value of κ determined using dressed-ion theory, which includes the effects of ion correlations [59]. Because the actual range of the potential is reduced relative to the mean-field prediction, the magnitude of the potential must increase to match the same $b_{22}$ value, which is why the fit values of *Z* and $W_{\mathrm{FUCHS}}$ are larger (see Figure 8c,d). With this correction, the increase in the $W_{\mathrm{FUCHS}}$ ratio closely matches the minimum aggregate growth observed for solutions with STPP concentrations ranging between 2.5 mM and 10 mM (which corresponds to ionic strengths between 25 and 100 mM). It is not possible to obtain accurate estimates for the FUCHS ratio at ionic strengths much greater than 100 mM because values of $b_{22\;}^{\mathrm{el}}$ become much less than the other contributions to $b_{22}$, which amplifies the uncertainty in the fitting parameter *Z*. However, it is not surprising that aggregate growth rates decrease at higher salt concentrations. Aggregate growth suppression occurs due to ion binding, which is saturable, while increased growth occurs due to ionic screening, which increases monotonically with salt concentration. The competition between these two effects gives rise to the minimum in the aggregate growth rates.

Another factor we have neglected is any ion pair formation between sodium and either ATP or TPP, which would cause the solution to be at a lower ionic strength. We expect this to be only a minor effect, as measured association constants are on the order of 10 M^−1^ [60,61], which corresponds to around 0.2% of the polyphosphate ions with bound sodium at a salt concentration of 10 mM.

For monoclonal antibody solutions, FUCHS ratios have been determined from fitting aggregate growth models to monomer loss data complemented with aggregate size measurements. The obtained values are on the order of 10^7^ to 10^9^, which is orders of magnitude greater than expected from DLVO theory indicating there are additional contributions to the stability of the aggregating units [53,55]. The high stability has been attributed to the low protein surface coverage of sticky protein aggregation hot spots, which lowers the likelihood that colliding units stick together. However, our analysis is still applicable if the primary effect of changing solution conditions on the FUCHS ratio is to alter the interaction potential, but not the distribution and nature of the aggregation prone regions.

### 3.8. ATP Attenuates Native-State Attractive Interactions between Ovalbumin Molecules

The discrepancy that aggregate growth rates are similar for ATP and TPP, while the measured values of $b_{22}\;$are different from each other, provides more insight into how the multivalent ions alter the protein–protein interaction potential. According to RLCA theory, a similar aggregate growth rate implies that the repulsive part of the interaction potential, which is predominantly determined by electrostatic interactions, is the same for ATP versus TPP solutions when compared at the same ionic strength. On the other hand, measured values of $b_{22}$ are much more sensitive to the contribution from short-ranged attractive interactions since the integral is related to the Boltzmann factor of the interaction potential. Taken together, these findings suggest that the difference between ATP and TPP is due to the effectiveness of ATP in attenuating short-ranged attractive interactions for native-state ovalbumin, rather than altering repulsive electrostatic interactions. Furthermore, at high ionic strength, where electrostatic interactions are sufficiently screened, the $b_{22}$ values for TPP-containing solution approach the curve to fit the NaCl data indicating similar attractive protein–protein interactions, while for ATP, $b_{22}\sim 1$ indicating very little protein–protein attraction. Because a constant short-ranged attractive contribution to $b_{22}\;$has been used when carrying out the fitting to determine $W_{\mathrm{FUCHS}}$, we expect the results obtained with STPP are more reliable, while the repulsive barrier will be over-estimated for the ATP solutions. We also note that the native-state attractive interactions, which are suppressed by ATP, are not involved in aggregate growth, which supports the general understanding that growth occurs by the addition of partially unfolded proteins [53,62].

The finding that the barrier in the interaction potential depends only on the electrostatic contribution has further implications for the nature of the attractive interactions. If attractive protein–protein interactions are isotropic, they should contribute to the barrier in the interaction potential at close separations where the electrostatic repulsion is greatest. This is especially evident when considering the integral in Equation (1) which is weighted by $r^{-2}$. The short-ranged interactions contribute much more to the FUCHS factor when compared against the integral for $b_{22}$ which is weighted by $r^{2}$. As such, it is much more likely the attractive interactions are anisotropic and constrained to occur in a limited fraction of the relative orientation space that defines the interaction between a pair of proteins. If this is the case, attractive interactions will only contribute a small fraction to the integral for the FUCHS factor. The deduction that attractive interactions are anisotropic is supported by computational modelling of protein–protein interactions [50,51,52,63,64] and phase behaviour studies [65,66,67], where the shape and location of phase boundaries can only be captured using anisotropic patchy models for describing proteins.

## 4. Discussion

By studying five different proteins, we hoped to gain an understanding of protein structural factors that determine whether ATP or TPP will be effective at reducing aggregate growth. The strongest suppression occurs with acidic proteins BSA or ovalbumin, where the initial increase in salt concentrations leads to immediate overcharging since the proteins already carry a net negative charge at neutral pH. However, the ζ-potential profile for α-Cgn is very similar to ovalbumin indicating a similar salt concentration is required to overcharge the protein. As such, it is unlikely the net charge of the protein is the only factor in determining the aggregation behaviour. For instance, RNaseA aggregate growth is suppressed, although the suppression occurs at higher salt concentration than observed for ovalbumin or BSA.

One factor that might play a role is the type of basic residues on the protein surface. A mutation study of the intrinsically disordered protein Histatin-5 involving all possible arginine-lysine swap mutants indicated TPP interacts more strongly with arginine versus lysine groups [27]. The arginine versus lysine mutants are more readily precipitated by TPP, which was attributed to non-specific TPP binding by arginine groups. Interestingly, the ζ-potential profiles as a function of TPP concentration are almost identical for all the mutants indicating TPP also preferentially binds lysine groups, but with a weaker binding affinity compared to arginine. Analogously, while non-specific binding of TPP to basic protein residues likely causes protein overcharging, we hypothesize that a requisite for electrostatic repulsion is a high content of arginine groups to tightly bind the multivalent ions. α-Cgn only contains four arginine groups which are distributed unevenly across the protein surface. However, it is unclear then why TPP does not significantly affect the aggregate growth of lysozyme, which is enriched in arginine groups. There is significant TPP overcharging of lysozyme by a salt concentration of 10 mM [30], but no aggregate growth suppression at concentrations of 30 mM and above, even though these concentrations are effective for RNaseA, ovalbumin and BSA. Lysozyme is precipitated at lower salt concentrations due to TPP-induced attractive protein–protein interactions. A distinct possibility is that these attractive interactions also exist at higher salt concentrations and balance any electrostatic repulsion arising from TPP overcharging lysozyme. Indeed, the measured $b_{22}$ values are less than −2 at pH 9 for the salt concentrations used in the aggregation studies shown in Figure 6 indicating sufficiently strong protein–protein attractions, albeit in the native state [30].

There has been much focus on determining the mechanism of how ATP alters protein self-assembly, aggregation, and fibril formation in vivo due to the mM concentrations of ATP occurring in intracellular environments. Our study supports the work by Mehringer et al. [26] who showed that TPP and ATP are equally effective at preventing aggregation of globular proteins, which we have attributed to a supercharging effect through tri-phosphate binding to basic residues of the protein surface. This mechanism alone cannot explain the behaviour in vivo since ATP, but not TPP, is effective at suppressing amyloid formation or solubilizing liquid condensates formed by proteins containing intrinsically-disordered regions. These effects have been attributed in part to the ability of the adenosine group to form π–π interactions with sticky aromatic groups [22,25,26,68]. However, we also expect electrostatic interactions to be a key factor in determining how ATP alters protein assembly in vivo since the screening length in the cytosolic environment has been estimated to be around 2.2 nm or an ionic strength of approximately 20 mM [69].

## 5. Conclusions

The non-specific nature of electrostatic interactions, where increasing protein net charge almost universally leads to a reduction in aggregate growth rates, suggests that using small multivalent ions to enhance colloidal stability could be an effective strategy for stabilizing a broad class of biotherapeutics. However, while the addition of the poly-phosphate ions always leads to an overcharging effect, this only translates to a reduction in aggregate growth rates for some of the proteins. As such we still require a better understanding of the mechanistic effects of the multivalent ions in order to rationalize their use. Key issues that need to be addressed are why overcharging by ions does not always lead to an increase in colloidal stability under native conditions, and whether increased colloidal stability due to overcharging always leads to aggregate growth suppression.

## Figures and Tables

**Figure 1 biomedicines-09-01646-f001:**
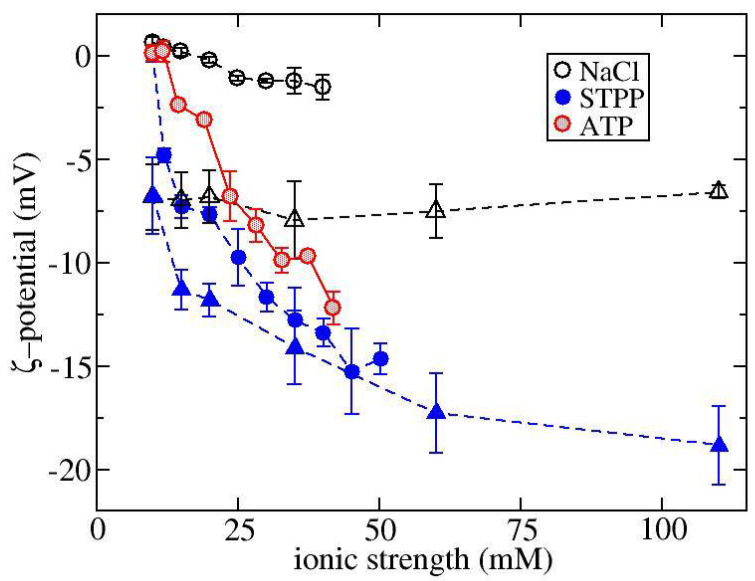
ζ-potential measurements as a function of ionic strength for solutions of ovalbumin (triangles) or α-Cgn (circles) with either NaCl, STPP, or ATP.

**Figure 2 biomedicines-09-01646-f002:**
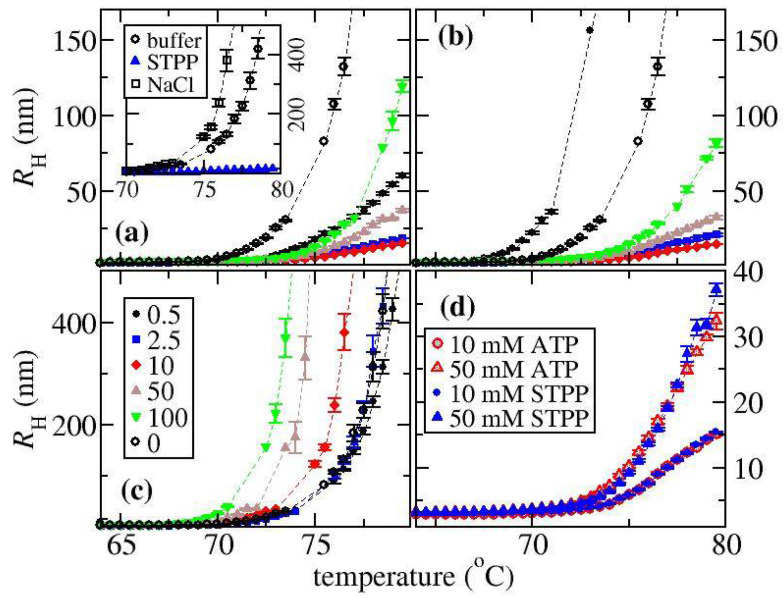
Measured values of $R_{H}$ determined from thermal ramps for solutions of ovalbumin containing different concentrations of (**a**) STPP, (**b**) ATP, (**c**) NaCl where the legend in (**a**) shows the concentrations of the excipients in mM. The inset to (**a**) contains a comparison of the buffer-only condition with samples containing either NaCl or STPP at a concentration of 10 mM. (**d**) corresponds to a comparison between solutions containing either ATP or STPP. Triplicate measurements were carried out for all samples with NaCl and all other conditions with ovalbumin were run in duplicate.

**Figure 3 biomedicines-09-01646-f003:**
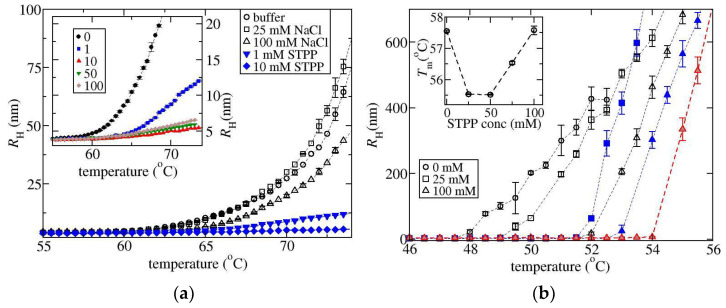
Measured values of $R_{H}\;$determined from thermal ramps for solutions containing (**a**) BSA with either STPP or NaCl and (**b**) α-Cgn with either STPP, ATP, or NaCl. Open, red-shaded, and blue-filled symbols correspond to NaCl, ATP, STPP, respectively. The inset to (**a**) corresponds to BSA in solutions with STPP where the legend denotes the concentration of STPP in mM. The inset to (**b**) is a plot of the melting temperatures measured for solutions of α-Cgn and STPP.

**Figure 4 biomedicines-09-01646-f004:**
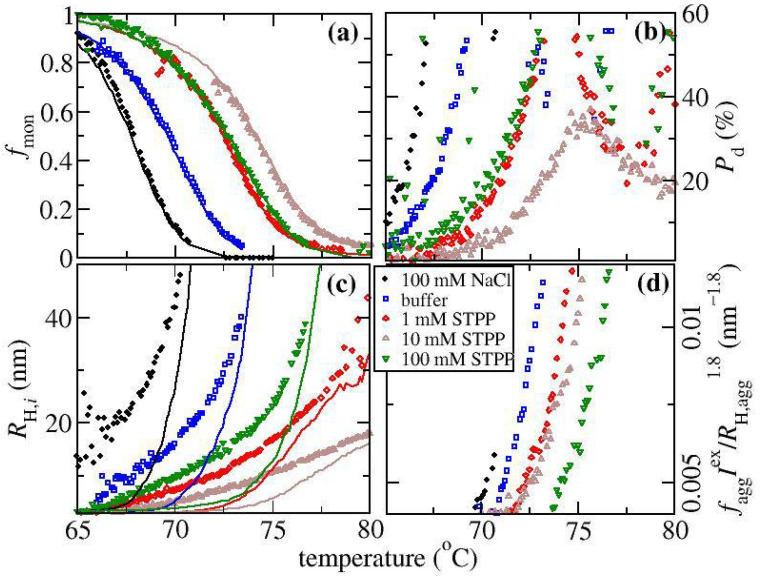
Results of thermal ramp experiments for ovalbumin solutions from the DLS analysis plotted versus temperature where salt conditions are shown in the legend. (**a**) The fraction of light scattering by the monomer fraction, $f_{\mathrm{mon}}$ determined from fitting two-decay model (symbols) and directly from the measured $I_{\theta }$ (lines). (**b**) The polydispersity $P_{d}$ determined from the cumulant analysis. (**c**) The hydrodynamic radius $R_{H,\mathrm{agg}}$ determined from the long-time mode of the two-decay analysis (symbols) and the $R_{H}$ values from the cumulant analysis (lines). (**d**) An estimate of the monomer concentration assuming aggregates have a fractal dimension equal to 1.8 (see text).

**Figure 5 biomedicines-09-01646-f005:**
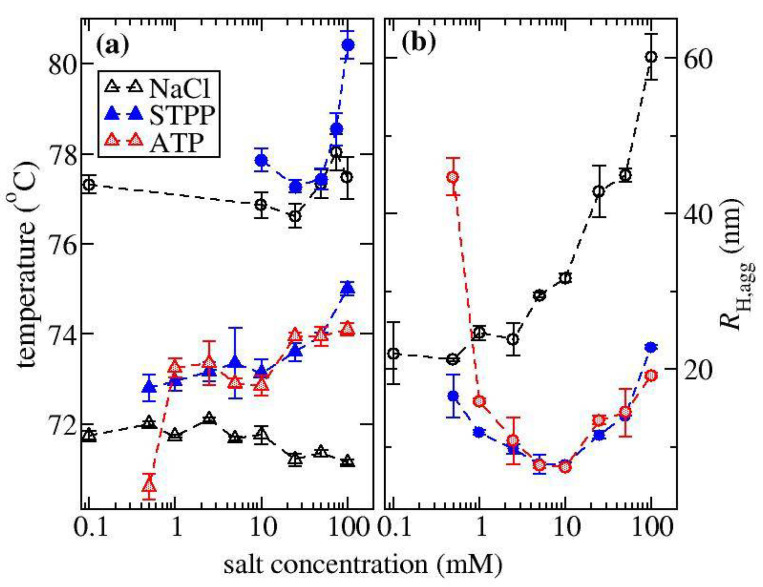
Results from thermal ramp experiments for ovalbumin solutions. (**a**) A plot of $T_{m}$ (circles), and the temperature where $f_{\mathrm{agg}}I^{\mathrm{ex}}/R_{H}^{1.8}=0.006$ nm^−1.8^ (up triangles) referred to as $T_{\mathrm{mon}}$. (**b**) The aggregate size $R_{H,\mathrm{agg}}$ as a function of salt concentration. Open, red-shaded, and blue-filled symbols correspond to NaCl, ATP, STPP, respectively.

**Figure 6 biomedicines-09-01646-f006:**
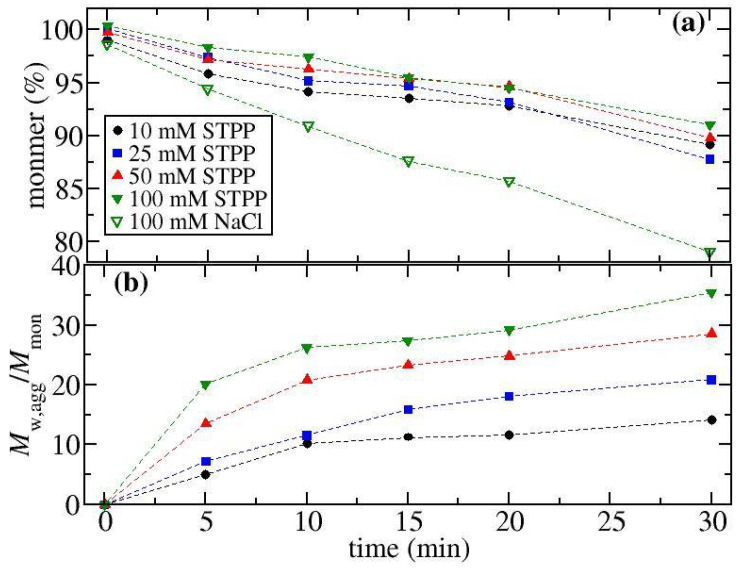
Plots obtained from the SEC-MALLS analysis of ovalbumin solutions containing either NaCl or STPP heated at 70 °C for different lengths of time. (**a**) Monomer remaining as a percentage of the initial monomer concentration plotted against time. (**b**) The weight average molecular weight of the aggregate peak obtained from the STPP samples plotted versus time.

**Figure 7 biomedicines-09-01646-f007:**
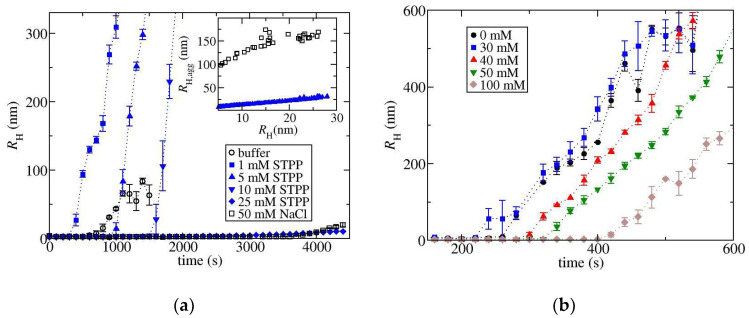
Results of isothermal aggregation studies for (**a**) RNaseA in solutions with either NaCl or STPP and (**b**) lysozyme for solutions with varying STPP concentrations, where the salt concentration is shown in the legend. Triplicate measurements were carried out for all lysozyme conditions. The inset to (**a**) contains $R_{H,\mathrm{agg}}$ determined from fitting to the two-decay model.

**Figure 8 biomedicines-09-01646-f008:**
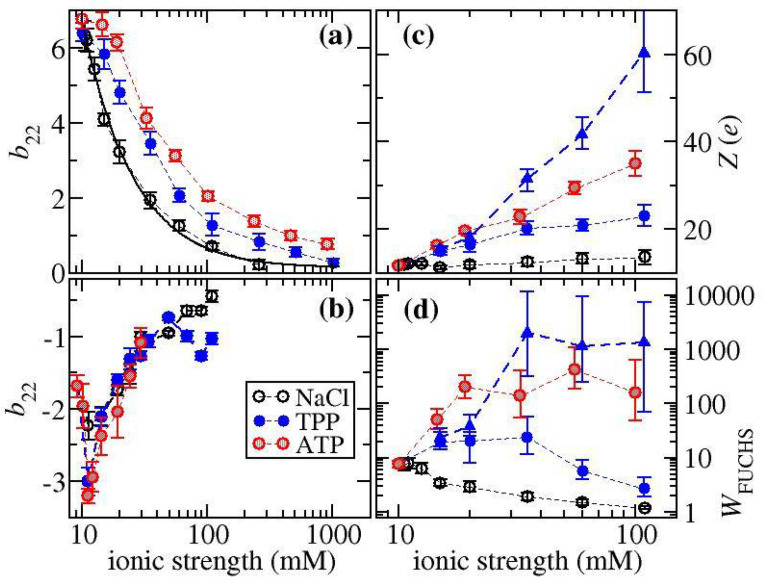
$b_{22}$ plotted as a function of ionic strength for solutions with either NaCl, STPP, or ATP and (**a**) ovalbumin or (**b**) α -Cgn. Symbols represent the experimental measurements while the solid line in (**a**) represents the best fit to the DLVO potential with the fit parameter Z=11.6 e. The values of (**c**) $W_{\mathrm{FUCHS}}$ and (**d**) Z were determined from fitting to each $B_{22}$ measurement for ovalbumin. For graphs (**c**,**d**), up triangles correspond to values obtained from fitting with a screening parameter κ that includes ion-ion correlation effects (see text for more details). Open, red-shaded, and blue-filled symbols correspond to NaCl, ATP, STPP, respectively.

## Data Availability

The data presented in this study are available on request from the corresponding author.

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
