# Peer review of "ATP and Tri-Polyphosphate (TPP) Suppress Protein Aggregate Growth by a Supercharging Mechanism"

_biomedicines, 2021, doi:10.3390/biomedicines9111646_

Round 1

Reviewer 1 Report

I have read with great interest the manuscript “ATP and tri-polyphosphate (TPP) suppress protein aggregate growth by a supercharging mechanism” by Bye et al and overall, I think it is a great and well-written study. I only have some minor concerns that are described below:

Figure 1: measurements for a-Cgn with ATP are missing, can the authors explain why?

Figure 2: I do understand that the range of the y axis is very different for NaCl compared to ATP and STPP because of the lack of protection against aggregation in increasing temperatures. However, it is kind of misleading the way it is presented as you cannot really correlate and compare the graphs between (a), (b), and (c). Perhaps the authors could split the y axis in (c) in two unequal parts, where the big one would go up to 150nm and the second smaller one up to 400nm. Additionally, in (d) it is quite difficult to distinguish the circles from the triangles or maybe it is just my age…

Figures 2 and 3: the description of the results for these figures in the text is very difficult to follow for several reasons. The ovalbumin data are presented first in Figure 2, while the BSA and a-Cgn data in Figure 3, but throughout the text they are not necessarily presented in the same order. Additionally, the BSA data are very briefly described in the text. What makes the situation even more complicated is the fact that the panels of the figures are almost not cited at all within the text and the reader needs to search in two separate figures the results while reading the text. Perhaps the authors should merge the two figures in one and they should definitely improve their text by citing the figure panels in the text.

Have the authors addressed the behaviour of disease-associated aggregation-prone proteins? For example, the natively unfolded protein tau is triggered to aggregate and form filamentous structures in vitro in the presence of polyanionic compounds, such as heparin, RNA, etc. I understand that the nature of such proteins is very different to ones tested here, but perhaps they could represent an interesting alternative to the currently presented mechanism and maybe the authors should at least discuss it.

Author Response

  1. Figure 1: measurements for a-Cgn with ATP are missing, can the authors explain why?

There is a mistake in the caption where the triangles refer to the ovalbumin data and the circles to the date for a-Cgn.  Due to time constraints, we had to be selective about what measurements done with STPP would be reproduced using ATP.  Given enough time, the z-potential measurements would have been carried out, but we thought that the  measurements for ovalbumin with ATP solutions provided enough evidence for the overcharging effect by ATP, which was the main point of the z-potential measurements.

  1. Figure 2: I do understand that the range of the y axis is very different for NaCl compared to ATP and STPP because of the lack of protection against aggregation in increasing temperatures. However, it is kind of misleading the way it is presented as you cannot really correlate and compare the graphs between (a), (b), and (c). Perhaps the authors could split the y axis in (c) in two unequal parts, where the big one would go up to 150nm and the second smaller one up to 400nm. Additionally, in (d) it is quite difficult to distinguish the circles from the triangles or maybe it is just my age…

We have included an inset to Figure 2a to provide a better comparison between the heating studies in STPP and in NaCl, which reflects the large difference in aggregate growth rates when adding the STPP.  Due to comments from referee 2, we have updated the figures so that less datapoints are shown.  We hope that it should now be easier to distinguish between the triangles and the circles in Figure 4d.

  1. Figures 2 and 3: the description of the results for these figures in the text is very difficult to follow for several reasons. The ovalbumin data are presented first in Figure 2, while the BSA and a-Cgn data in Figure 3, but throughout the text they are not necessarily presented in the same order. Additionally, the BSA data are very briefly described in the text. What makes the situation even more complicated is the fact that the panels of the figures are almost not cited at all within the text and the reader needs to search in two separate figures the results while reading the text. Perhaps the authors should merge the two figures in one and they should definitely improve their text by citing the figure panels in the text.

We have reorganized Section 3.2 to make the description of the thermal ramp experiments easier to follow.  The first paragraph summarized all the experiments and introduces the figures.  Each of the following paragraphs is then dedicated to describing the behaviour of each protein separately in the order of ovalbumin, BSA, and then a-Cgn.  We have also referred to the graphs in the text more often to guide the reader.

  1. Have the authors addressed the behaviour of disease-associated aggregation-prone proteins? For example, the natively unfolded protein tau is triggered to aggregate and form filamentous structures in vitroin the presence of polyanionic compounds, such as heparin, RNA, etc. I understand that the nature of such proteins is very different to ones tested here, but perhaps they could represent an interesting alternative to the currently presented mechanism and maybe the authors should at least discuss it.

This is a very good point, although we are not so familiar with the amyloid aggregation literature.  In the discussion we have pointed out that we cannot rationalize the effect of ATP binding on amyloid formation because ATP behaves differently from TPP, and we can only say that we expect electrostatic interactions will be significant under low salt conditions, but how they impact the behaviour is difficult to ascertain due to the differences in rate determining steps for amyloidogenic proteins versus natively-folded protein aggregation.  We are aware of studies showing small poly-phosphate ions increase amyloid formation of various intrinsically disordered proteins, but without further reading, it is not clear how to interpret those findings in terms of the overcharging mechanism discussed in our paper. As such, we prefer to avoid discussing this topic, but will consider it for the future.

Reviewer 2 Report

The mechanism of ATP and TPP reducing protein aggregation is an interesting topic.

Some comments:

  1. The title is “ATP and trip-polyphosphate (TPP) suppress protein aggregate growth by a supercharging mechanism”. However, this is not for all protein as reported in the manuscript, such as α-Cgn. So the title might need some modification to match your results.
  2. It is important to repeat each experiment to confirm the accuracy of your data. However, the authors only reported repeated experiment for ζ-potentials measurements. This causes some concerns. For example, in figure 2c, the line of measured RH value determined from thermal ramps for solution containing 0 NaCl is different from the line for 0 STPP and 0 ATP in Fig. 2a and 2b, corresponsive. All three lines contained RH value measured for same condition (ovalbumine at buffer solution), and were supposed to be similar. The inconsistence will raise the question about the reliability of your data. The repeated measurements from one sample can’t be considered as repeated experiments. The mean of the repeated measurements from one sample should be used and considered as one data point. Multiple independent experiments need to be carried out to be analyzed and reported/plotted in mean±SD in the figures. Please clarify this in the method section.
  3. Please add the unit of y-axis in Fig. 4d.
  4. In page 8, paragraph 2, last sentence: the authors concluded that “The greatest shift to higher temperatures occurs for the conditions with either 5 or 10 mM STPP, while…..”. However, no data for 5 mM STPP was reported in Fig. 4a. Is this a typo error, or is it a data not shown? Please clarify.
  5. Similarly, in P9, paragraph 1, line 10-11: “When fmon…., which remains low for the condition with 5 to 25 mM STPP”. No data was shown for the conditions containing 5 and 25 mM STPP in Figure 4. Please clarify.
  6. P9, paragraph 1, line 8-9: “The peak in Pd occurs at a temperature….. With further increasing temperature Pd reduced since….”. However, in Fig. 4b, it looks like there is a increase for 1 and 100 mM STPP when temperature passed about 76 degrees, but this is not seen for 10 mM STPP. What is the possible explanation of this?
  7. P9, paragraph 2, line 12-14: “For the runs with 10 mM STPP, results are only obtainable once the temperature is greater than ~72.5°C”. There are data obtained at the temperature lower that that for 10 mM STPP as shown in Fig. 4d. Is “10 mM STPP” an typo error, and it should refers to 100 mM STPP as shown in fig. 4d.
  8. 5a: What is the data of Tm measured under the condition containing STPP lower than 10 mM. It looks like Tm would be higher at the lower STPP concentration. Without the data, it would be not proper to conclude that “the values of Tmon is parallel to the changes in Tm” as stated in the text. Also the value of Tmon is much lower at ~0.6 MM STPP while the value of Tmon is much stable at higher salt concentration. Please explain what caused this difference? Also did authors measure Tm for ATP with different concentration? That will give a more full understanding of how ATP affect ovalbumin solution.
  9. The authors cited previous research that indicate monomer loss kinetics under thermal stress correlate with the proteim Tm. However, Fig. 5b is not fully correlate to Fig. 5a. There is a shift of salt concentration from ~25 mM to 5-10 mM STPP corresponding to the minimum Tm and RH,agg. Do you have any explanation about what causes this shift?
  10. Figure 8: the figure legend is not matching the figures for of Fig. 8c and 8d. Please correct it.
  11. P12, paragraph 3, line 1-2: “The behavior for…. in solutions a few pH units removed from the PI,….” The sentence is kind of confusion. Do you mean “… in solutions a few pH units lower than the PI…”? Please clarify this.
  12. P12, paragraph 3, line 5-7: The authors stated that “When changing the salt from NaCl to STPP, the increasing b22 values parallels the increase in the ζ-potential magnitude, ….”. I will not agree with the word “parallels”. For ζ-potentials, the increase become greater with higher ionic strength as shown in Fig. 1. While for b22 valuse, the magnitude keeps the same at >20 mM ionic strength as shown in Fig. 8a.
  13. For result session 3.9, both FUCHS factor and b22 are not show good correlation with aggregate growth rats. As shown in Fig. 5b, RH,agg shows a V shape for both STPP and ATP, and is increasing from 10 mM to 100 mM for both STPP and ATP. But WPUCHS only shows increase at low concentration, and then reach plateau (ATP) or even shows decrease (STPP solution) as shown in Fig. 8d. Please re-write the title and context.
  14. The first several paragraphs of discussion section are like a continuing discussion following the result of the result session 3.9. Please re-write or re-organize the text.
  15. P16, paragraph 1, last sentence: “For instance, RNaseA aggregate growth is suppressed, although at higher salt concentration than occurs for ovalbumin or BSA”. This sentence is not a completed sentence. Please re-write it.
  16. Typo error:

In P9, paragraph 1, line 3: “Figure 3c” should be Figure 4c.

In P9, paragraph 1, line 4: “Figure 3b” should be Figure 4b.

In P11, paragraph 2, line 3-4: “Figure 6A and 6B” should be Figure 7a and 7b.

In P11, paragraph 2, line 16: “see inset to Figure 6a” should be “see inset to Figure 7a”.

In P12, the figure legend in this page should be “Figure 7” instead of “Figure 6”.

In P14, eqation 3: (b22)el should be (b22)att.

In P16, paragraph 3, line 16: please change “RNase A” to “RNaseA”.

  1. Figure S2 of the SI: please provide the value of x-axia in the figures.
  2. There is some issue with Figure S3 of the SI. Please replace it with the correct ones.

Author Response

We thank both referees for careful reading of the manuscript and their insightful comments, which we believe will make the manuscript much improved.  We have provided responses to all the comments.  Referee 2 was critical and suggested that major revisions were required.  We believe this was due in part to questions about the experimental design, which is addressed in detail in the rebuttal to comment 2 below, and the inability of our model for the FUCHS factor to capture the patterns of ovalbumin aggregate growth, which is addressed in the rebuttal to comment 13 below.  

  1. The title is “ATP and trip-polyphosphate (TPP) suppress protein aggregate growth by a supercharging mechanism”. However, this is not for all protein as reported in the manuscript, such as α-Cgn. So the title might need some modification to match your results.

We don’t believe the title suggests ATP or TPP suppresses aggregate growth for all proteins, in this case, we would have used the word “universal” in the title.  In addition, in the abstract, we have made it clear that aggregate growth suppression only occurs for some proteins, not all.  However, we are open to changing the title if the editors and/or the referee feel strongly that the title is mis-leading.

  1. It is important to repeat each experiment to confirm the accuracy of your data. However, the authors only reported repeated experiment for ζ-potentials measurements. This causes some concerns. For example, in figure 2c, the line of measured RHvalue determined from thermal ramps for solution containing 0 NaCl is different from the line for 0 STPP and 0 ATP in Fig. 2a and 2b, corresponsive. All three lines contained RHvalue measured for same condition (ovalbumin at buffer solution), and were supposed to be similar. The inconsistence will raise the question about the reliability of your data. The repeated measurements from one sample can’t be considered as repeated experiments. The mean of the repeated measurements from one sample should be used and considered as one data point. Multiple independent experiments need to be carried out to be analyzed and reported/plotted in mean±SD in the figures. Please clarify this in the method section.

The data shown in Figure 2c for 0 mM NaCl is indeed the same data as shown in Figure 2a and 2b, but the scale is different in Figure 2c, which might have led to the confusion.  Most of the thermal ramp experiments were carried out in duplicate measurements.  The exceptions are the thermal ramp with ovalbumin in NaCl, which were carried out in triplicate.  These measurements were carried out first, because they showed very good reproducibility, we opted to carry out measurements in duplicate and believe the results are reliable, especially considering the patterns we observe with changing solution conditions (eg results for ovalbumin with STPP and ATP are almost identical to each other).  We have followed the suggestion to include error bars on our measurements. In Figures 2, 3, and 6, we provide the average across the replicate runs and include the standard deviation as the error bar (noting that the standard deviation for two datapoints represents the range of the data).  We have also reduced the number of datapoints by averaging over small temperature or time intervals to make the plots easier to follow.  The experiments with lysozyme were also carried out in triplicate as the runs exhibited more variation compared to the other proteins. 

We also note that we only carried out single timepoints for the SEC-MALLS runs shown in Figrue 6, which are only used to show that aggregate sizes are much smaller in STPP than in NaCl solutions, and that aggregate sizes decrease when increasing STPP concentration above 10 mM.  We believe the data support this assertation without requiring replicate measurements due to the clear trends in molecular weight observed with increasing STPP concentration.

  1. Please add the unit of y-axis in Fig. 4d.

We have added the units to the y-axis of figure 4d, we also pointed out on page 9 that  corresponds to a normalized excess light scattering intensity so that the property will be dimensionless.

  1. In page 8, paragraph 2, last sentence: the authors concluded that “The greatest shift to higher temperatures occurs for the conditions with either 5 or 10 mM STPP, while…..”. However, no data for 5 mM STPP was reported in Fig. 4a. Is this a typo error, or is it a data not shown? Please clarify.

The analysis has been carried out for all the runs with ovalbumin, but it is not possible to show all the results in the figure.  However, we have used the analysis to deduce the parameters whon in Figure 5, which indicates that the 5 and 10 mM runs are very similar to each other.  Because this data has not been shown in Figure 4 though, we have updated the statement on page 8 to indicate the greatest shift occurs for the 10 mM run.

  1. Similarly, in P9, paragraph 1, line 10-11: “When fmon…., which remains low for the condition with 5 to 25 mM STPP”. No data was shown for the conditions containing 5 and 25 mM STPP in Figure 4. Please clarify.

Here we have included a statement that the data is only shown for 10 mM STPP.

  1. P9, paragraph 1, line 8-9: “The peak in Pdoccurs at a temperature….. With further increasing temperature Pdreduced since….”. However, in Fig. 4b, it looks like there is a increase for 1 and 100 mM STPP when temperature passed about 76 degrees, but this is not seen for 10 mM STPP. What is the possible explanation of this?

We have included a line to indicate that growth is likely occurring by aggregate-aggregate coalescence under these conditions. In general, both growth mechanisms will be occurring and the relative contribution of the different mechanisms is not only determined by the solution conditions, but also the aggregation time, after longer periods of time there will be a lower monomer population and higher aggregate concentration so that the relative contribution from coalescence versus growth by chain polymerization will increase with time (at least for short times).  Discriminating between these details requires much more detailed kinetic studies, eg monomer loss kinetics, which is beyond the scope of our current study.    

  1. P9, paragraph 2, line 12-14: “For the runs with 10 mM STPP, results are only obtainable once the temperature is greater than ~72.5°C”. There are data obtained at the temperature lower that that for 10 mM STPP as shown in Fig. 4d. Is “10 mM STPP” an typo error, and it should refers to 100 mM STPP as shown in fig. 4d.

We have removed this statement, we can only obtain reliable estimates for the mass protein concentration above a temperature at which the aggregate size is large enough for the aggregate morphology to be characterized by the same fractal dimension.

  1. 5a: What is the data of Tm measured under the condition containing STPP lower than 10 mM. It looks like Tm would be higher at the lower STPP concentration. Without the data, it would be not proper to conclude that “the values of Tmonis parallel to the changes in Tm” as stated in the text.

We agree with the referee that we cannot make the conclusion without the  data at low salt concentrations.  Another study (reference 26) has shown that the  for ovalbumin in solutions with either STPP or ATP is constant between 1 mM and 50 mM salt concentration, while there is a slight increase at 1 mM salt concentration relative to the buffer solution.  We have reworded the paragraph and included the reference.

Also the value of Tmon is much lower at ~0.6 MM STPP while the value of Tmon is much stable at higher salt concentration. Please explain what caused this difference? Also did authors measure Tm for ATP with different concentration? That will give a more full understanding of how ATP affect ovalbumin solution.

We believe the referee is referring to the 0.6 mM ATP solution.  We could not measure the melting temperatures in ATP solutions by intrinsic fluorescence since ATP absorbs light at the wavelengths used in the experiment. However, as mentioned in the previous comment,  has been measured by differential scanning calorimetry for ovalbumin in ATP and in STPP containing solutions in reference 26.  The parameters are very similar to each other.  We do not have the data for 0.5 mM ATP, so it is not possible to explain the abnormal behaviour at this solution condition. However, we believe this is an important result and could be of interest so that we choose not to exclude this result. 

  1. The authors cited previous research that indicate monomer loss kinetics under thermal stress correlate with the proteim Tm. However, Fig. 5b is not fully correlate to Fig. 5a. There is a shift of salt concentration from ~25 mM to 5-10 mM STPP corresponding to the minimum Tmand RH,agg. Do you have any explanation about what causes this shift?

  1. Figure 8: the figure legend is not matching the figures for of Fig. 8c and 8d. Please correct it.

We have updated the figure legend to indicate results for ATP are also shown.

  1. P12, paragraph 3, line 1-2: “The behavior for…. in solutions a few pH units removed from the PI,….” The sentence is kind of confusion. Do you mean “… in solutions a few pH units lower than the PI…”? Please clarify this.

For ovalbumin this behaviour is actually occurring a few pH units above the pH, not below, since ovalubumin pI is below 5.  However, as alluded to by the referee, there are examples of proteins with basic pIs, including lysozyme or mAbs, where the behaviour occurs a few pH units below the pI.  As such, because many of the studies examining electrostatic interactions between proteins (references 31, 46-50) involve basic proteins, we have updated the sentence to indicate above or below the pI to avoid confusion.

  1. P12, paragraph 3, line 5-7: The authors stated that “When changing the salt from NaCl to STPP, the increasing b22 values parallels the increase in the ζ-potential magnitude, ….”. I will not agree with the word “parallels”. For ζ-potentials, the increase become greater with higher ionic strength as shown in Fig. 1. While for b22 valuse, the magnitude keeps the same at >20 mM ionic strength as shown in Fig. 8a.

We agree that this statement is mis-leading.  We have expanded this point to indicate that the increase in the  value at fixed ionic strength suggests that there is an increase in the fixed charge on the protein that arises from ion binding, which is consistent with the ζ-potential measurements.

  1. For result session 3.9, both FUCHS factor and b22 are not show good correlation with aggregate growth rats. As shown in Fig. 5b, RH,aggshows a V shape for both STPP and ATP, and is increasing from 10 mM to 100 mM for both STPP and ATP. But WPUCHS only shows increase at low concentration, and then reach plateau (ATP) or even shows decrease (STPP solution) as shown in Fig. 8d. Please re-write the title and context.

We disagree with this statement.  It is important to note that the ionic strength for STPP solutions is ten times the salt concentration.  As such, an ionic strength of 100 mM corresponds to a salt concentration of 10 mM.  After correcting for the effect of ion-ion correlations on the screening length, we find that the fuchs factor is significantly enhanced for the salt concentrations of 2.5 mM to 10 mM (ionic strengths of 25 to 100 mM – see triangle symbols in Figure 8d).  This corresponds to the minimum in our aggregate growth parameter shown in Figure 5b which occurs also between 2.5 and 10 mM.  We have included a statement on page 15 to emphasize this point.  Unfortunately, it is not possible to extend our analysis to higher salt concentrations because the range of the electric double layer force is reduced so it does not make as significant a contribution to  in which case there is a very large uncertainty when discriminating between the electrostatic term and the other contributions from short-ranged attractions and excluded volume to .  Hopefully this point is now more clear in the revised version as it may have been difficult to follow in our original submission

We note that in the article we have covered the literature studies indicating that aggregate growth of ovalbumin under the low salt conditions is controlled by electrostatic repulsion (references 41 to 43).  In addition, for the NaCl solutions, we have shown the reduction in aggregate growth does correlate with the screening of electrostatic repulsion which is reflected by the  values.  In addition, we have covered literature which has shown that aggregate growth under conditions corresponding to high  values is controlled by electrostatic interactions (references 14, 15, 17).  Next, our combined z-potential and  measurements indicate that TPP in particular causes an increase in electrostatic repulsion through overcharging the protein.  Because aggregate growth of ovalbumin is controlled by electrostatic interactions under these conditions, it follows that overcharging is the likely cause of the alteration in aggregate growth rates.  The only question that remains is why do the aggregate growth rates not correlate with the  values in STPP or ATP solutions, and we have addressed this point by showing the FUCHS factor, which is more directly related to aggregate growth rate, does reflect the significant reduction in aggregate growth observed in solutions with 2.5 mM to 10 mM (figure 8d) 

  1. The first several paragraphs of discussion section are like a continuing discussion following the result of the result session 3.9. Please re-write or re-organize the text.

We have included a separate section in the results part for the first two paragraphs of the discussion to better reflect that the paragraphs are based upon considering the results shown in Figure 8.

  1. P16, paragraph 1, last sentence: “For instance, RNaseA aggregate growth is suppressed, although at higher salt concentration than occurs for ovalbumin or BSA”. This sentence is not a completed sentence. Please re-write it.

The sentence has been updated.

  1. Typo error:

We sincerely apologize for these typographical errors (which arose when we changed the figure numbering system) and regret the difficulties that they have caused for the referees in reading the manuscript. 

In P9, paragraph 1, line 3: “Figure 3c” should be Figure 4c.

In P9, paragraph 1, line 4: “Figure 3b” should be Figure 4b.

In P11, paragraph 2, line 3-4: “Figure 6A and 6B” should be Figure 7a and 7b.

In P11, paragraph 2, line 16: “see inset to Figure 6a” should be “see inset to Figure 7a”.

In P12, the figure legend in this page should be “Figure 7” instead of “Figure 6”.

In P16, paragraph 3, line 16: please change “RNase A” to “RNaseA”.

In P14, eqation 3: (b22)el should be (b22)att.

This is not a typographical error, Equation 3 is the starting point to determine the parameters describing the electric double layer force, which are then used for calculating the Fuchs ratio.  The attractive contribution to  is highly anisotropic and cannot be written in terms of an integral over a centrosymmetic interaction.  However, we showed that the repulsive contribution, because it is longer ranged and less anisotropic, can be written in terms of Equation 3 (for instance see reference 31).  In a recent study we have also shown such a simple model can capture the concentrated solution thermodynamic properties of a weakly interacting antibody solution (see Lanzaro et al (2021) molecular pharmaceutics v18: 2669).

  1. Figure S2 of the SI: please provide the value of x-axia in the figures.

Figure S2 has been updated

  1. There is some issue with Figure S3 of the SI. Please replace it with the correct ones.

Figure S3 has been updated